# Artificial Intelligence: A Promising Tool in Exploring the Phytomicrobiome in Managing Disease and Promoting Plant Health

**DOI:** 10.3390/plants12091852

**Published:** 2023-04-30

**Authors:** Liang Zhao, Sean Walkowiak, Wannakuwattewaduge Gerard Dilantha Fernando

**Affiliations:** 1Department of Plant Science, University of Manitoba, Winnipeg, MB R3T 2N2, Canada; 2Canadian Grain Commission, Winnipeg, MB R3T 6C5, Canada

**Keywords:** taxonomic and function annotation for microbiome sequencing, synthetic microbial communities (SynComs), microbe–plant association, artificial intelligence, machine learning, disease forecasting

## Abstract

There is increasing interest in harnessing the microbiome to improve cropping systems. With the availability of high—throughput and low—cost sequencing technologies, gathering microbiome data is becoming more routine. However, the analysis of microbiome data is challenged by the size and complexity of the data, and the incomplete nature of many microbiome databases. Further, to bring microbiome data value, it often needs to be analyzed in conjunction with other complex data that impact on crop health and disease management, such as plant genotype and environmental factors. Artificial intelligence (AI), boosted through deep learning (DL), has achieved significant breakthroughs and is a powerful tool for managing large complex datasets such as the interplay between the microbiome, crop plants, and their environment. In this review, we aim to provide readers with a brief introduction to AI techniques, and we introduce how AI has been applied to areas of microbiome sequencing taxonomy, the functional annotation for microbiome sequences, associating the microbiome community with host traits, designing synthetic communities, genomic selection, field phenotyping, and disease forecasting. At the end of this review, we proposed further efforts that are required to fully exploit the power of AI in studying phytomicrobiomes.

## 1. Introduction

In natural growth environments, plants interact with diverse microorganisms such as bacteria, fungi, oomycetes, archaea, and viruses [1]. In fast—changing and stressful conditions, plants associate tightly with these microbes in various ways, such as nutrient uptake, plant growth and development, and plant health fitness [2]. Rather than functioning and evolving independently, plants and their associated microorganisms exist cooperatively as part of a biological system. Additionally, the plant—associated microbiome, also known as the phytomicrobiome, is essential to the plant life—cycle. When facing stress from insects and/or pathogens, plants can recruit protective microorganisms to suppress the invasive agents. Microbial communities can promote resistance and tolerance to adverse abiotic stress conditions, such as heat, drought, or high salinity [3,4]. In parallel, microbial communities can also be affected by the plant through molecular interactions [5].

A deep understanding of the interactions between the plant and its microbiome make it possible to leverage microbiome information in sustainable crop production. As biotic stresses (e.g., plant pathogen and insect) and abiotic stresses under the current climate change trend are ever—increasing threats to agricultural production, there is an urgent need to reduce the usage of pesticides and agrochemicals [6,7], and to promote crop production and enhance resistance against stresses through manipulating the plant—associated microbiome [8].

However, the complex interactions between plants, their pathogens, soil microbiota, and environmental conditions under natural field conditions makes it challenging to predict disease development and plant health. It has been difficult to understand stochastic and deterministic drivers from multiple factors including the genetic background of the host [9] and pathogen [9], the density dynamics of the pathogen population [10], phytomicrobiome composition and assembly [11], and temporal and geographic differences in stress [12]. Even for agricultural environments under relatively similar stochastic and temporal conditions, considerable variations in disease dynamics exist [13,14]. This indicates that the interactions between plants and the phytomicrobiome can be largely determined by local agricultural environments, thereby raising a series of questions such as: (1) “which microbes exist in the phytomicrobiome?”, (2) “which biological roles or functions from microbes or communities are beneficial to pathogen repression and plant growth promotion?”, (3) “how can we associate the microbiome with host traits and disease development and use this knowledge to design microbe community?”, (4) “how can we accelerate the breeding progress to increase plant resistance to disease?”, (5) “how can we effectively conduct disease phenotyping in the field?”, and (6) “How can we forecast plant disease based on existing knowledge on field microbiome, plant genome, weather conditions, etc.?”.

Understanding plant–microbe interactions and answering the above—mentioned questions require information such as the microbes at different taxonomy levels, metagenome and transcriptome data, the functions of whole microbe communities, and the impacts of synergistic factors on plants [3]. In evaluating crop phenotypes such as disease development, stress resistance, and physiological status, this information needs to be integrated in order to precisely and comprehensively understand these traits [15,16]. With a deep understanding, plant health under ever—changing environmental stresses can be managed in a better way [17]. Computational methods could help us to understand plant–pathogen interactions and the roles of plant microbiomes.

Artificial intelligence (AI) is a general discipline that focuses on understanding and developing systems that display intelligence properties, which are capable of generalizing and deriving knowledge from existing information [18]. Recent advancements in this area are contributed to by a subset of AI techniques, and machine learning (ML) and its sub—family, deep learning (DL), through which models learn patterns from large amounts of raw datasets. AI models have been developed to understand microbiome data because of their powerful ability in handling diverse multivariate data. Here, we review the AI models and their application in microbiome data analysis, and describe the applications of these models in phytomicrobiome studies. In the following sections, we will first introduce AI algorithms. Then, we will review key questions in microbiome studies and how AI models can be applied to these tasks. The main objectives of this review are, first, to introduce AI and its sub—discipline concepts, second, to identify the major limitations of traditional bioinformatics methods on the microbiome data analysis, and third, to highlight AI features that may help to circumvent these limitations.

## 2. Major AI Models

AI has proven capable of improving performance in many areas such as predicting earthquakes, classifying plant species based on leaf/plant images, automobile self—driving, recognizing faces, and filtering emails. ChatGPT, the most advanced natural language processing (NLP) AI model [19], received more than 100 million users within the first two months and receives 13 million visits a day as of 2023. Even in an editorial article, ChatGPT was used to generate content to introduce a special journal issue to illustrate how advances in AI can help stem cell researchers [20]. Another example to show the power of AI comes from the AlphaFold, which predicts the three—dimensional protein structure based only on its amino acid sequence [21]. AlphaFold can regularly predict protein structures with competitive accuracy to experimental structures in most cases, and greatly outperformed other computational methods in the 14th Critical Assessment of Protein Structure Prediction.

ML (a subset of the AI technique) refers to a diversity of methods that analyze data statistically, which has the ability to adapt and improve model performance [22]. In other words, ML infers knowledge from existing data, applies the knowledge to new data, and makes corresponding predictions. ML can be generally grouped into unsupervised and supervised learning. Unsupervised learning enables users to explore unidentified patterns and to cluster unlabeled datasets without human intervention. In contrast, supervised learning focuses on interpolating the patterns for a labeled dataset, and needs assistance from human experience or knowledge.

### 2.1. Unsupervised ML

Unsupervised ML methods can be applied to two types of tasks: (1) clustering tasks that group data on the basis of data similarity, and (2) dimension reduction tasks, which regenerate representative features from a large number of variables. One method for clustering analysis is k—means (Figure 1A). The objective of the k—means method is to group *n* observations into a specific number of (k) non—overlapping clusters based on distance calculation, with each data point belonging to only one cluster.

Dimension reduction involves taking observations from high—dimensional variables and transform them into low—dimensional features to compress a large number of variables into a limited number of features while retaining useful information and minimizing information loss [23]. In dimension reduction tasks, principal component analysis (PCA) is most commonly used [24]. In PCA, a set of uncorrelated principal components (PCs) are converted from the original variables (Figure 1B). PCs contain the most important information that explains the variance of the original observations. In addition, PCs reduce the number of variables that represent the original observation to a small number of PCs instead of the large set of original variables; this can simplify linear or logistic regression analyses. Principle coordinate analysis (PCoA) is a similar method to PCA but it uses dissimilarity measures rather than correlations. PCoA is commonly adopted in microbiome studies. For example, PCs generated from operational taxonomic units (OTUs) from microbiome information were visualized and prepared for further analysis [25,26].

### 2.2. Supervised ML

Unlike unsupervised methods, supervised ML can be more elaborate in phytomicrobiome data analysis, such as the taxonomic annotation of sequence data, gene function analysis, or trait–microbiome association. In supervised learning, each microbiome dataset is grouped into a specific category and assigned a label (Y). During the training process, models adjust their parameters to minimize the difference between their calculated outcome and the ground truth. Once trained, the adjusted parameters can be functional for extracting information from unlabeled datasets and assigning labels (Y). Among the supervised learning approaches, some commonly used algorithms will be introduced in the following sections, including support vector machine (SVM), random forest (RF) [27], artificial neural network (ANN), and DL—based methods.

SVM [28] is a widely used classification method. SVM searches for a non—linear or linear separating surface from the provided dataset to maximize the distance to the nearest training variables of designated labels (Figure 1C). A decision boundary is drawn to separate each class while maximizing the margin from the nearest samples. As SVM is effective in high—dimensional data and is computationally tractable, it is well—suited to microbiome data. For example, SVM was successfully used to predict agricultural soil health using microbiome composition generated from 16S rRNA gene sequences in a continental—scale study [29].

The RF comes from the concepts of decision trees and bootstrap aggregation (bagging) [30]. Each decision tree starts with a basic question to separate data entries, followed by other questions that are added step—wise (Figure 1D). Each question helps a data entry to reach a final decision. In processing data, hundreds or thousands of decision trees can be constructed in an RF model. Bagging, another feather of RF, integrates close trees by choosing a specific or average value. Through this operation, RF is an ideal algorithm to identify a “real outcome” in complex and heterogeneous data, which is common in microbial datasets, and thus it is widely used in a diversity of classification tasks that involve high—dimensional microbiome data [31]. The high—dimension microbiome data usually have limited numbers of samples and could result in model overfitting problems, which happens when the model contains too many parameters than what is needed to justify the data, and may fail to predict future observations reliably. The RF algorithm is less impacted upon by the overfitting problem, as RF is made up of subunits that are trained completely independently on subsets of the training data. Thus, it is appealing in microbial community analysis. The RF models can also estimate the prediction power for each variable and thus provide meaningful information in analyzing the relative importance of each of the factors [32].

Mimicking the neurons in a brain, ANN employs a network of artificial neurons to solve learning problems [33]. ANN contains many weights. The training process adjusts weights based on differences between the ground truth and the generated output. The model structure and dimension can be adjusted by adding or deleting hidden layers. Thus, ANNs are very flexible in handling complex and high—dimensional datasets, making them a powerful technique in analyzing the role of microbes in complex settings [34]. In addition, ANNs do not involve data processing, dimension reduction, or feature selection in conducting classification tasks. However, as with ANNs, it is difficult to trace how a decision is made, and they are often regarded as ‘‘black box” approaches.

DL, as a much more advanced form of ANN, is gaining more attention in microbiome studies. Different from original ANNs, DL consists of many hidden layers [35,36]. In this way, the multiple hidden layers can reveal non—linear relationships between the input and the output data [37], and thus, they can perform very complex functions that are insensitive to background noise and sensitive to minute but informative signals.

### 2.3. Major Deep Learning Architectures

There are several major DL architectures, including convolutional neural networks (CNNs), recurrent neural networks (RNNs), and transformer. CNNs were originally developed to perform tasks whose input variables are distributed in a space pattern, such as 2D/3D images or sequential information such as sounds, texts, DNA sequences, or single nucleotide polymorphisms (SNPs) (Figure 2A). CNN models contain at least one convolutional layer in their architectures [38,39]. In a convolutional layer, the convolutional operation is conducted with a pre—defined window (height × width) and stride across the input data matrix. One of the main advantages of CNNs is their capability to capture space features [40]. At the shallower or beginning layers, more basic and low—level features can be learnt, while the learnt features become more specific and descriptive with the depth of the network. The convolutional layers thus act as automatic feature extractors from the input matrix.

RNNs are specifically designed to receive sequential inputs such as DNA sequences (the data are composed of nucleotide-by-nucleotide), texts (the data are composed of word-by-word), and audios (the data are composed of note-by-note). Unlike ANN and CNN, which process all input units at the same time, RNN processes the inputs progressively one-by-one, based on input sequence (Figure 2B). In the RNN, the output from previous neuron(s) is processed at the current neuron, together with the current input, making the current neuron produce an output with the consideration of “past memories”. RNNs are very useful in handling sequence data, but are hard to train because of the vanishing/exploding gradient problem [42]. Long short-term memory (LSTM), a special type of RNN algorithm, has been developed to solve this problem. LSTM can memorize long-term dependent information through the input gates, forget gates, and output gates, which finally control the information that needs to be forgotten and passed to the next unit [43].

Transformer architecture consists of input positional encoding, and encoder and decoder parts. A positional encoding tensor assigns a relative position to each input unit. Thus, the input units can be differentiated based on their positions in the sequence, even when some of them might be the same in meaning [41]. The encoder part includes self-attention, feed-forward networks and layer normalization (Figure 2C). A self-attention mechanism was originally proposed for natural language processing, to extract features from an extremely long-distance crossing of 10,000 input units with parallel computing [41]. The self-attention mechanism can directly acquire long-distance dependencies for any combination of positions in a sequence data, rather than traversing all the positions from itself to its dependencies, such as what should be achieved in RNN or LSTM. Thus, the self-attention mechanism improved the capability for extracting information from sequence data. In a transformer, the self-attention module calculates the attention score for all nucleotides in the DNA reads with respect to a specific one, and thus it could learn the pairwise relationships between nucleotide sequences. The related nucleotides will be designated with a high probability of relationship with respect to the focused ones and create a combined representation. When these nucleotide sequences are combined, the effects of these nucleotides will be high and most of the other irrelevant nucleotides can be filtered out.

## 3. AI Applications in Taxonomic Annotation, Gene Function Annotation, Associating Plant Traits, and Designing Synthetic Microbe Communities

Microbial studies have benefited from the advancement of the sequencing technique, where the technological revolution has increased the DNA sequencing length and has reduced the cost when sequencing a large number of microbe samples. The major sequencing technique used in plant microbiome studies includes 16S amplicon sequencing, internal transcribed spacer (ITS) sequencing, and metagenomic shotgun sequencing. 16S amplicon sequencing amplifies a conserved region of the 16S rRNA gene, which is found in all bacteria and archaea, and the resulting data can be used to identify the types and relative abundances of different microorganisms in the community [44]. ITS sequencing specifically, on the other hand, amplifies the ITS region of eukaryotes, which is highly variable among different fungal species, making it a useful marker for identifying and characterizing fungal communities [45]. Metagenomics shotgun sequencing is a more comprehensive approach that can be used to study the diversity of all microorganisms, including bacteria, archaea, fungi, viruses, and even small eukaryotes, in a sample [46]. It involves randomly fragmenting the DNA extracted from the sample and sequencing the resulting fragments. The resulting data can be used to reconstruct the genomes of the organisms present in the sample, and to identify the types and relative abundance of different microorganisms in the community. Each of these techniques has its advantages and limitations. The technique of 16S amplicon sequencing is specific to bacteria and archaea, but it can be less informative at the species level and may not capture the full diversity of the community [44]. ITS sequencing is highly specific to fungi and can provide a detailed view of fungal diversity, but it provides less reliable information about other microorganisms present in the sample [45]. Metagenomics shotgun sequencing can provide information about all microorganisms in the sample, but it can be more complex and expensive to perform and analyze [46]. The choice of technique depends on the research question and the specific microbial community being studied.

A diversity of bioinformatics tools has been designed to classify the DNA sequences in the metagenome into taxonomic and functional groups. There are two major annotational approaches. The first one consists of k—mer–based approaches including CENTRIFUGE [47], GeneMark [48], and kraken2 [49]. A k—mer is a substring of length k that occurs within a DNA sequence, and k—mer–based methods involve counting the frequency of the occurrence of each k—mer within the sequence data to generate a k—mer frequency table. This table can be used to identify regions of the genome that are likely to be protein—coding genes, non—coding RNA genes, repetitive sequences, or other functional elements. One limitation of the k—mer based method is that it can be computationally intensive, particularly for large datasets or complex genomes. Generating a k—mer frequency table requires counting of the occurrence of every possible k—mer within the sequence data, which can be time—consuming and memory—intensive. As a result, some k—mer–based methods may not be practical for use with very large datasets. Another limitation is that if there are errors or gaps in the sequence, this can affect the accuracy of the k—mer frequency table and may lead to errors in annotations. Additionally, k—mer–based methods are not effective at identifying genes or functional elements that have low k—mer frequencies or that are otherwise difficult to detect based on k—mer content alone.

The second class of algorithms are the alignment—based algorithms which classify sequences based on sequence similarity with reference genomes in the databases. These types of methods include DIAMOND [50] and BLAST [51]. These methods involve aligning the query sequence to a reference database to identify the similarity of the query sequence to the reference. Alignment—based methods have several advantages over k—mer–based methods. They can be more effective at identifying genes or other functional elements. They can also be more accurate in cases where the query sequence is highly divergent from the reference sequence or where there are significant structural differences between the two sequences. However, this method can be computationally intensive. Additionally, alignment—based methods rely on the accuracy and completeness of the reference database. Thus, annotating DNA sequences into the levels of species, taxonomy, functions, pathway and host trait is challenging, as not all reference genomes have been annotated into a similar range of ranks and the annotation ability becomes lower with the increasing annotation specificity.

In recent years, ML— and DL—based algorithms have been developed to annotate sequencing data. ML—based methods are capable of learning complex patterns in the sequencing data that may be difficult to detect using other methods. This can be particularly useful for identifying non—coding functional elements or for predicting the functions of novel genes [52]. ML algorithms can also be adapted to different types of sequence data and annotation tasks, making them versatile tools for genome annotation and functional characterization [53]. Handling large datasets with many variables is another advantage of ML—based methods, making them well—suited for analyzing complex genomic data. However, ML methods usually require large, high—quality datasets for model development, and the accuracy and generalizability of the predictions can be influenced by the quality and representativeness of the training data. Additionally, ML methods require significant computational resources and expertise in machine learning and bioinformatics. Overall, machine learning methods are a valuable tool for genome annotation and functional characterization, particularly when combined with other annotation methods to achieve comprehensive and accurate results. For example, general linearized models are commonly used to differentiate the microbial composition of samples, while PCA was applied to decreasing data dimension and data visualization [54]. Learning and predicting the health status of plants with metagenome samples is not common, but an insightful study has been conducted for clinical metagenomics studies with more than 2400 metagenome samples [55]. In this section, we will attempt to illustrate ML and DL applications in phytomicrobiome sequence analysis, and we have summarized some of the related applications in Table 1.

**Table 1 plants-12-01852-t001:** Summary of AI techniques used in microbiome-based analyses. This table briefly summarizes AI applications in the areas of microbe taxonomic annotation, function annotation, association with host traits, and designing SynComs. This table is not exhaustive but it mentions current and commonly employed methods that are tailored for microbiome data or specific tasks.

Reference	Research Type	Research Priority	Raw Sequences	AI Method
[56]	Taxonomic analysis	Assignment of raw sequences to the origin of the genome without knowing the reference genome.	Microbiome datasets from several different habitats from GMGCv1 (Global Microbial Gene Catalog), including the human gut, non-human guts, and environmental habitats (ocean and soil).	Semi-supervised learning
[57]	Taxonomic analysis	Annotation of viral components in mixed metagenomes containing both viral and host contigs.	Sequences subsampled from prokaryotes and viral genome sequences at several contig lengths: 500, 1000, 3000, 5000, and 10,000 bp.	Unspecified machine learning
[58]	Taxonomic analysis	Taxonomic identification of microbial eukaryotes from integral components of natural microbial communities.	Raw sequence reads of microbial samples mainly originating from groundwater.	SVM
[59]	Taxonomic analysis	Classification of microbes into species and genera, and the estimation of abundance for human gut microbiomes.	2505 representative genomes of human gut microbe species.	LSTM; self-attention
[60]	Taxonomic analysis	Identification of eukaryotic sequences in metagenomic datasets.	Datasets from NCBI and the Joint Genome Institute, including 8220 genomic sequences representing Eukarya (4381) (nuclear (73), plastid (2260) and mitochondrial genomes (2048)), Bacteria (1860), and Archaea (1979).	ANN
[61]	Taxonomic analysis	Identification of phage sequences without a reference genome.	Metagenomic sequences from NCBI.	LSTM
[62]	Functional annotation	Prediction of antibacterial or antifungal activity based on features of known natural product biosynthetic gene clusters.	Biosynthetic gene clusters that were available from the Minimum Information about a Biosynthetic Gene Custer database (version 1.4).	SVM; RF
[63]	Functional annotation	Functional annotation and classification of the complete (genomic proteins) and partial (metagenomic ORFs) protein sequences.	Protein sequences and associated information of orthologous groups of genes (from eggNOGv3.0).	RF
[64]	Functional annotation	Identification of biosynthetic gene clusters in bacterial genomes, and improved identification precision and ability to identify novel functional gene classes.	Open reading frames in 3376 reference bacterial genomes.	BiLSTM
[65]	Functional annotation	Identification of transcription activator-like effector that causes bacterial leaf streak of rice.	Promoter sequences, defined as the 1000 bases upstream of the start codon, for the approximately 56,000 rice genes annotated in the MSU Rice Genome Annotation Project Release 7.	Naive Bayes and logistic regression classifiers
[66]	Functional annotation	Identification of promoters in atypical microbial hosts.	Promoter sequences from the *Geobacillus* 7544 core coding sequence.	RF, ANN, and partial least squares regression (PLS)
[67]	Functional annotation	Annotation of *Lactococcus* genes with molecular functions needed for biological nitrogen fixation in Sierra Mixe maize, including mucilage carbohydrate catabolism, glycan-mediated host adhesion, iron/siderophore utilization, and oxidation/reduction control.	Whole genome sequences.	RF
[68]	Functional annotation	Identification of bacterial sequence functions that are associated with the growth of the plant *Brassica rapa* in different soil microbial treatments and at different stages of plant development.	16S rRNA amplicon variants.	Generalized linear and Bayesian multilevel modeling
[69]	Functional annotation	Annotation of DNA sequences of crop pathogens for functions in nutrient acquisition, avoidance of host defenses, regulation of symbiosis, symbiosis, and movement in the environment of another organism.	16S rRNA amplicons.	SVM; RF
[70]	Functional annotation	Classification of non-ribosomal peptides from soil-associated microbes with a high tolerance to sequence modification.	The DNA sequences of microbial datasets from *Xenorhabdus* and *Photorhabdus* families (XPF), *Staphylococcus* (SkinStaph), soil-dwelling *Actinobacteria* (SoilActi), and a collection of soil-associated bacteria within *Bacillus*, *Pseudomonas*, *Buttiauxella*, and *Rahnella* genera generated under the Tiny Earth antibiotic discovery project (TinyEarth).	SVM
[71]	Functional annotation	Annotation of non-ribosomal peptides.	Nucleotide sequences including complete and draft genome assemblies.	SVM
[72]	Functional annotation	Identification of potential sources of novel antibiotic resistance genes (ARGs).	ARG genes were obtained from three major databases: CARD, ARDB, and UNIPROT.	ANN
[73]	Functional annotation	Gene prediction using metagenomics fragments.	Sequencing reads from Orphelia and MGC metagenomic dataset.	CNN
[74]	Association with host traits	Rice traits (dried biomass, tissue nitrogen concentrations, and net photosynthetic rate) were associated with bacterial microbiota, including those in the seed, root endosphere, and rhizosphere.	16S rRNA amplicons.	RF
[75]	Association with host traits	Classification of fungi into lifestyle classes (pathogen, saprobe, or others).	The whole genome of 101 *Dothideomycetes.*	SVM
[16]	Association with host traits	Association of crop productivity with bulk soil microbiome composition and several nitrogen utility-related taxa.	Shotgun sequences for bulk soil samples.	RF
[76]	Association with host traits	Association of rhizosphere microflora and root exudate profiles to cucumber resistance to Fusarium wilt disease.	16S rRNA amplicons.	PCA; RF
[77]	Association with host traits	Association of a microbiome profile with its original location.	Microbiome datasets (16S rRNA amplicon and shotgun sequences) from Boston urban and blinded samples from eight cities.	RF
[78]	Association with host traits	Association of root microbiomes with rice traits, including sulfur oxidation and reduction, biofilm production, nitrogen fixation, denitrification, and phosphorus metabolism.	16S rRNA amplicons.	RF
[79]	Association with host traits	Association of the relative OTUs abundance with rice age, and identification of OTUs in the rhizosphere and endosphere compartments that discriminate rice age.	16S rRNA of 1510 samples from root spatial compartments in field-grown rice (*Oryza sativa*) throughout three consecutive growing seasons, as well as two geographic sites.	PCoA, RF
[80]	Association with host traits	Association of root microbiota with rice developmental stages.	16S rRNA amplicons.	RF
[81]	Association with host traits	Association of root microbiota with different *Panax* species.	Amplicon sequencing for 405 multi-niche samples of three Holarctic distinct *Panax* species.	RF
[82]	Association with host traits	Revealing worldwide soil microbial community patterns by merging independent taxonomy-based data sets.	16S rRNA amplicons.	RF
[83]	Association with host traits	Deciphering the functional relationship between soil-specific microbes and ecosystem properties.	16S rRNA amplicons.	Neural network; RF
[84]	Designing SynComs	Development of a novel approach to design microbe communities and to predict plant response to phosphate starvation.	16S rRNA amplicons	ANN

### 3.1. AI Applications in Taxonomic Analysis

ML algorithms can be trained to conduct taxonomic analysis based on their genomic data and other phenotypic information. This can be particularly useful for identifying and characterizing novel organisms. For example, McHardy et al. (2007) first applied a SVM model to classify assembled metagenomic contigs into different taxonomic ranges [85]. The authors evaluated the SVM prediction ability, based on the class output, ranging from genus to domain taxonomic levels. The sensitivity of this SVM model reached 0.9 for long DNA sequences. The model was further developed into a web-based tool for annotating metagenomes by Patil et al. (2012) [86]. The SVM model was also developed recently by Vervier et al. (2016) [87] to provide a flexible taxonomic annotation function. As certain alignment-based classifiers could not classify sequences into different taxonomic levels, this SVM model presented a rank-flexible method that could output the most appropriate level to classify the sequences based on the maximum score across all of the different rank-orientated models.

Fiannaca et al. (2018) introduced DL methods of CNN and ANN to annotate 16S sequences and to classify metagenomics data [34]. The sequence data were processed into a digital image format and then fed into the DL models. These two models were compared to the baseline 16S ribosome database project (RDP) classifier, which is the Naive Bayesian (NB) classifier [88]. The results indicated that the CNN and DBN models (reaching 91.3% in accuracy) outperformed the NB classifier (83.8% in accuracy). Liang et al. (2020) developed the DeepMicrobes model, which involves two DL algorithms, bidirectional LSTM, and a self-attention (which is a basic block for the transformer) model, to classify metagenomic reads [59]. To optimize models, the authors tested approximately 30 parameters such as the input data encoding method, different DL algorithms, and the inclusion of a self-attention mechanism. Through comparison, the bidirectional LSTM model outperformed the other models. The model was then compared with the state-of-the-art classifiers, CLARK-S [89], Centrifuge [47], kraken2 [90], and Kaiju [91]. In general, the DeepMicrobes model outperformed the other annotation tools in the perspectives of recall, precision, and genus abundance estimation. Nevertheless, kraken2 and Kaiju performed better in estimating abundance at the species level, although the annotation ability was relatively lower.

### 3.2. AI Applications in Functional Annotation

Functional information and its potential role in the habitat environment of the microbial community can also be explored with AI. Examples of studies that have applied ML to microbial functional analysis are listed in Table 1. Sharma et al. (2015) developed a two-step annotation process named WOODS [63]. The first step is to classify sequencing reads with ML, followed by the second step, alignment-based annotation. In this method, the ML functions as a pre-processing step to align the fragments to a gene category. The RAPsearch2 tool [92] was used in the alignment step, and eggNOG3 [93] was used as the functional reference. Among the ML models evaluated in this study, RF outperformed other models in classifying the testing dataset, and the overall function alignment performance was better than via BLAST. As the related ML application in phytomicrobiome analysis was not found to our best effort, we tried to use an ML application for detecting the ARG from metagenomic data. As microbial resistance is becoming a widespread concern, the screening of ARGs in microbial communities received a lot of attention recently [94,95]. Detecting AR genes directly from shotgun metagenomic data is very challenging. Limited by the related reference databases, alignment methods were less effective compared to the ML-based methods. Arango-Argoty et al. (2018) developed DeepARG, a DL-based method, to detect 30 categories of AG determinants in metagenomic sequences [72]. For this method, the accuracy of AR gene annotation outperformed alignment-based tools, although DeepARG also takes alignment scores as inputs. The better performance might be explained by the principle that DL adapts the threshold automatically in identifying AR categories during the learning process, rather than manually designating a similarity threshold for each AR class.

An integrative framework has also been developed to detect functional genes from microbial communities. This framework uses ML-based models and analyzes the data combining microbial composition and context information such as the structures of phylogenetic relationships in the communities [96]. This model was designed to detect OTU features and to predict their functions. With a CNN-like DL model, Khodabandelou et al. (2019) illustrated the promise of DL in annotating microbe sequences for their function [97]. This model is capable of annotating short sequences with functions such as promoters for different species. Al-Ajlan and El Allali (2019) developed another CNN model (CNN-MGP) to detect genes from metagenomic DNA reads with no pre-selection for the features [73]. Candidate fragments can be selected with a preset possibility value of 0.5 with the following post-processing manipulation. With this DL method, functional annotation can be applied in a set of reference databases, including KEGG, GenBank, COG [98], FunGene [99], and MG-RAST [100].

The annotation process, which mainly relies on the connection between the DNA sequence and the function of the encoded protein, can be impacted upon by the confidence threshold for the similarity between the query and the reference sequence. When annotating the functions of sequencing reads, different thresholds need to be assigned based on the nature of the function in traditional methods. However, as different reference databases have their own specificities, there is no such threshold standard to annotate function [93,101,102,103,104,105,106]. Such diversity in annotation standards makes it challenging to transfer knowledge from one to another dataset. AI, the method of which is capable of transferring knowledge through training with a dataset but applied to another dataset, might be useful for conducting functional annotation with better performance, although this has not been used commonly in handling phytomicrobiome data. This method is particularly useful for when there are undefined microbe species in the reference dataset.

### 3.3. AI Application in Plant–Microbiota Association Analysis

A set of core microbiome members could be used to determine the key functions of a microbiome community and to speculate plant traits such as plant resistance to biotic/abiotic stresses [107] and productivity [108]. However, it is the whole community rather than several core taxa that enable observable effects on the plant [109]. In natural environments, the plant trait is the overall outcome of the plant genome [110], characteristics of stress [111], soil environments [112], the microbe community, and other undiscovered players. A minor difference in microbe constitution or function, such as nitrogen-fixing ability, pathogenicity, or toxicity may have a subtle influence on plant traits [111]. Therefore, it is difficult to attribute a microbiome with specific plant traits with classical methods, particularly when some species in the microbiome are unclassified.

Recent advances in ML methods make it possible to predict plant traits directly from the overall microbiome data. We summarized some of the ML applications in microbe–plant associations in Table 1. A phytomicrobiome association with the plant host involves other environmental conditions such as temperatures, soil conditions, organic matter, nutrition, or metal ions. As such, when considering the interaction between the plant and the environment, one must also consider environmental differences. This may limit the applications of certain MLs and DLs, which may predict poorly across different environments. Nevertheless, a number of datasets were constructed in research projects to associate microbiome data and plant traits such as drought stress [112], plant disease [107,113], and crop productivity [16,108]. Chang et al. (2017) used the RF approach to classify and separate plant productivity into low or high categories based on identified soil microbial taxa information [16]. The relative contribution of each category can be estimated to reflect sample productivity. DL methods have also been introduced in predicting host phenotype in this end-to-end way [16].

### 3.4. AI Design of Synthetic Communities

The phytomicrobiome can promote plant growth or repress pathogens. Changing the microbiome via artificial inoculation with a community of plant growth-promoting rhizobacteria (PGPRs) can benefit host growth, control pathogens, and resist abiotic stresses. Some PGPRs mediate plant hormone production such as cytokinins, gibberellins, and auxins, while other PGPRs such as *Bacillus* spp., *Arthrobacter* spp., and *Pseudomonas* spp. are able to produce 1-aminocyclopropane-1-carboxylic acid (ACC) deaminase [114] to reduce stress symptoms under adverse environmental conditions such as drought. N fixation, phosphate solubilization, auxin production, and other pathways can be induced with PGPRs such as *Pseudomonas* spp., *Pantoea* spp., and *Paraburkholderia* spp. to promote soybean or wheat growth through the enhancement of stress resistance and nutrient uptake [114]. However, these functions are not achieved by a single microbe, but instead by complex interactive microbial communities. The concept of synthetic microbial communities (SynComs) has been proposed to solve this problem. SynComs refers to a small-scale consortia of microbe that are artificially designed to apply the acquired knowledge of function and structure from the microbiome in natural environments. The principle of SynComs is to preserve the native interactions between microbes and plants while reducing the complexity of the microbial community. By doing so, the key functions of the microbial community can be achieved and manipulated practically.

Manipulating SynComs is an advanced technology to discover microbe–plant associations. By removing and adding microbe members in SynComs, the function of each member can be identified. For example, removing *Enterobacter cloacae* from the SynComs helped in discovering its function in mitigating maize blight disease [115]. Using an ANN model, Herrera Paredes et al. (2018) developed a novel approach to designing microbe communities and predicting the plant response to phosphate starvation [84]. This approach studied microbe–plant bilateral interactions to infer the causal relationships between microbiota memberships and host phenotypes in phosphate accumulation. In the study, a set of partially overlapped SynComs were defined and their effects on plant phenotype were tested, and plant response at the transcriptional level was also analyzed. Through evaluating the performance of different models, an ANN model was selected to conduct plant phenotype prediction because of its better performance compared to the two linear models. Strikingly, 23 out of 25 ANN-guided designs were validated through experimental assays, supporting the advantage of AI methods in assisting SynComs design. However, designing SynComs with AI have been rarely reported. Although this technique has been reported as a promising technique in reviews, we did not find other AI studies in SynComs except the above-mentioned one, reflecting the infant stage of AI in this area.

## 4. AI Applications in Plant Genomic Prediction against Pathogens, Phenotyping, Plant–Microbiome Interactions, and Disease Forecasting

### 4.1. AI Applications in Genomic Prediction against Pathogens

Disease resistance can be qualitative or quantitative, which can often be attributed to differences in the plant genome. The scope of qualitative disease resistance is generally conditioned for by a single resistance (R) gene recognizing avirulence factors in a classic gene-for-gene mechanism, and the inheritance is said to be qualitative or Mendelian. In contrast, quantitative resistance is usually conditioned by many genes of small effect, and the inheritance is said to be quantitative or polygenic [116]. For qualitative traits that are controlled by single genes, DNA markers are often used to screen and to select the desired gene in breeding programs through marker-assisted selection (MAS). The identification of accurate markers that are strongly associated with the trait is the key to the successful application of MAS. However, this will be more challenging for complex traits controlled by many genes. An alternative method that investigates quantitative traits is genomic selection or genomic prediction (GP). GP takes advantage of all molecular markers, regardless of the significance threshold, to determine the breeding and/or genetic potential of a candidate individual for selection. Compared to MAS, GP has several advantages. GP allows breeders to select individuals with desirable traits at an earlier stage of the breeding process, which can save time and resources. At the same time, GP can reduce the cost of selection by reducing the need for phenotyping, which can be expensive and time-consuming. More importantly, GP can be used to select complex traits that are difficult to measure directly, such as disease resistance and yield.

The key element in GP is to build a robust and accurate statistical model based on available individuals with both phenotypic and genotypic data. Statistical models have been developed to improve the robustness and prediction accuracy. One of the frequently used models is the genomic best linear unbiased prediction (GBLUP). It was built based on the assumption that all SNPs contribute to the heritability of breeding traits and that they rise from the same normal distribution [117,118]. However, this assumption could reduce the prediction ability of the linear mixed model (LMM) when the trait under study is controlled under several dominant genes. In fitting these effects, Bayesian-based methods have been developed, including BayesA, BayesB, BayesLASSO, etc. [119,120,121,122,123]. The Bayesian-based methods assume that SNPs belong to different groups that have their own independent variances and specific distributions, such as the inverse chi-squared distribution. Although these traditional LMM or Bayesian-based approaches have been used in plant breeding, they are developed based on the assumption that genotype random effects follow a prior distribution and that each genotype contributes to the associated phenotype independently. Such assumptions require a large number of samples to dilute the effects of population structure. At the same time, the individual genotype effect may not follow a specific distribution perfectly. Additionally, these approaches are all based on a linear mapping from genotype to phenotypes, and it is less powerful for them to capture non-linear effects such as dominance and epistasis, which are common and important in complex traits [124,125].

To overcome the limitations of assumptions about the genetic architecture and the linear effects, ML approaches were developed. These methods do not require pre-assumptions and they are capable of extracting non-linear features. Many of these methods have been applied in GP problems, including but not limited to SVM with non-linear kernels (i.e., radial basis function SVRrbf and polynomial SVRpoly [126,127], reproducing kernel Hilbert spaces (RKHS) regression [128,129], and Gradient Tree Boosting (GTB) [130], as well as RF [130,131]).

DL is regarded as an efficient method in several studies of GP [130,132,133,134,135,136,137,138,139] because of its capability in handling a diversity of high-dimensional tasks [134,140]. After major innovations in recent years, advanced DL architectures have been developed to conduct complex trait predictions in several crops [141,142]. Jubair et al. (2021) developed a transformer-based DL model, GPTransformer, to conduct GP for barley resistance to Fusarium head blight (FHB) which is caused by *Fusarium graminearum* Schwabe [143]. Two pathogen inoculation methods were used to fully explore the possible pathogen–plant interactions. The first method inoculated the barley plant with the microbe communities on maize kernels which were infected with two strains of *F. graminearum*. In GP, the pre-screened essential genomic markers were fed into a GPTransformer to predict FHB and deoxynivalenol (DON). The results indicated that the GPTransformer performed similarly to the GBLUP model, with only 1% improvement over BLUP for DON and the performance for FHB. This study suggests the potential of DL methods in understanding pathogen–plant interactions or predicting plant disease phenotype when compared to the popular BLUP model.

However, ML models did not always outperform other methods for all traits and species. Linear models tend to perform consistently across predictions, while the ML models varied substantially from trait to trait. Montesinos-López et al. (2018) compared three DL architectures of ANN, CNN, and RNN against the commonly used linear GBLUP model with nine datasets [144]. Generally, GBLUP achieved the best performance in eight out of nine datasets when considering the interaction between the genotype and the environment. Interestingly, DL outperformed GBLUP in six out of the nine datasets when ignoring the interactions. From a larger view, Montesinos-López et al. (2021) surveyed 23 papers and found that no relevant differences in prediction performance were found between DL methods and the conventional linear models [145]. Specifically, DL performed better in 11 out of the 23 studies when taking into account the interaction between genotype and environment interaction, while 13 of these studies observed a better performance of DL when ignoring the genotype × environment interaction.

One of the reasons for the modest performance of DL could be that the number of training samples for most GP tasks was not sufficient for DL to learn non-linear interactions when the number of SNPs (or background SNPs) is too large. It is particularly so under a flawed experimental design which failed to screen out noisy SNPs, and the traits have major effect loci. With datasets containing different numbers of SNP markers from six plant species, Azodi et al. (2019) found that non-linear methods showed better performances in predicting traits in the datasets containing fewer markers. Reasonably reducing background information could improve DL performance. Pook et al. (2020) added a convolutional layer to intensify information, which were then fed into the ANN layers (referred to as LCNNs) [146]. In this way, the model performance was improved significantly compared to ANNs, regardless of data size. It is interesting to note that adding a convolutional layer to intensify information does not involve human screening of the markers, which is an advantage of the DL models in refining maker information over statistical methods.

DL’s ability in handling GP tasks can also be improved by adapting advanced DL architectures. In most cases, CNN-based models are more advanced in capturing spatial information and can therefore outperform the relatively simple ANN models when they are compared together. For example, using the International Maize and Wheat Improvement Center (CYMMIT) datasets, Pérez-Enciso and Zingaretti (2019) benchmarked several ANNs and CNNs [147]. It was found that CNNs always outperformed ANNs in GP. Similarly, in the study conducted by Ma et al. (2018), by investigating 2000 wheat lines and 33,000 markers, CNNs showed a much better prediction performance than ANN models [148]. Further, it seems that LSTM is more appropriate for handling sequential data and for exploring SNP dependencies. Maldonado et al. (2020) exploited the potential of LSTM architecture in conducting GP on *Zea mays* L. and *Eucalyptus globulus* Labill [149]. A significant increase in prediction performance was observed in LSTM compared to the other ML method, linear models of GBLUP, and different types of Bayesian regression models. On the contrary, when the CNN architecture was compared against linear Bayesian models, its GP performance was less attractive on polyploid outcrossing species of strawberry and blueberry [150]. The different performances of CNN and LSTM compared to conventional methods in the above-mentioned two studies may be attributed to the genome differences, presence of interactions, sample size, or model tuning. However, the differences in architecture strength between CNN and LSTM cannot be ignored, although it is still hard to make a conclusion because of the limited LSTM applications in GP tasks. As we introduced before, CNN is better at feature extraction from 2D data, while RNNs (or LSTMs) are more advantageous for sequential data. The nature of SNPs is a series of mutants on a genome sequence, and their sequential property and SNP dependencies might more easily captured by LSTM models.

A deeper understanding of both the DL architectures and the biological questions is also important in constructing DL networks. For example, given the situation where adjacent SNPs usually have no underlying direct functional relation, region-specific filters were introduced by adding a local CNN layer to reduce the background noise, and the GP performance was improved significantly [146]. As DL models are not outstanding for all applications, they can be integrated with conventional ML and/or linear models. For example, Jeong et al. (2020) integrated four types of models of CNN, RF, DNN, and RRB into the GMStool to conduct GP tasks [151]. As these individual models could capture SNP features from different aspects, the GMStool achieved the best prediction performance on the testing dataset. In addition, the microbes associated with plant growth environments are critical to disease development, and GP performance is difficult to improve if the variance and composition of microbe communities are ignored. Unfortunately, to our knowledge, no study has integrated phytomicrobiome information into consideration when conducting GP. Along with phytomicrobiome data accumulation and DL method improvement, it is very promising to improve crop trait prediction.

### 4.2. High-Throughput Plant Disease Phenotyping and In-Field Plant Disease Forecasting

High-throughput phenotyping is another trend for assisting with the discovery of pathogen–plant interactions in the field. Rather than using quantitative or binary phenotypes, other formats of phenotypes can be recorded using the formats of images, videos, or even sounds in the AI models. Several efforts, from local to international, are ongoing in order to construct phenomics centers for plant pathogen studies, which automate and standardize high-throughput measurements of plant phenotypes at all levels [152]. The physiology, development, and growth, as well as other traits of plants, can be recorded in a fast, non-invasive, and less costly strategy [153] using AI techniques. DL has been widely used to classify and detect various diseases [154,155,156,157]. The recognition and classification of maize leaf diseases, including northern corn leaf blight (*Exserohilum*), common rust (*Puccinia sorghi*), and gray leaf spot (*Cercospora*) diseases have been conducted using DL with an accuracy of 93.35% [158]. In cucumber (*Cucumis sativus*), a semantic segmentation model based on CNN was developed to segment powdery mildew disease on leaf images at the pixel level, and the pixel accuracy of the CNN model (96.08%) was higher than the segmentation methods of K-means, RF, and GBDT [159]. In pearl millet (*Pennisetum glaucum*), DL has been applied for the identification of mildew disease, and an accuracy of 95.00% was reported for the developed model [160]. Our lab is also developing DL models to effectively rate canola blackleg disease and flea beetle bites on seedling cotyledons. At the same time, disease progression can be acquired with hyperspectral imaging techniques [161] and plants metabolite evaluation can be measured with mass spectrometry (MS) and nuclear magnetic resonance (NMR) spectroscopy techniques [162], while plant transpiration and temperature can be measured through infrared thermography [162]. All the information can be fed into the DL model for analysis. In addition, to improve phenotype accuracy, proteomics, microbiome, metabolomics, imaginary data, weather, and soil data can all be integrated. Processing these complex and high-dimensional data is an ideal scenario for advanced DL architectures such as CNN, LSTM, or Transformer because of their excellent abilities in dealing with images, sequential data, and heterogeneous datasets with multiple variables and outcomes. As the DL-based high-throughput phenotyping technique has been reviewed [163,164,165], there is no need to review this technique explicitly in this section.

Multi-lateral interactions among microbe, plant and environment make it very challenging to make predictions on disease outbreaks in the field. In natural environments, several factors such as microbe community, pathogen genome, plant genome [110], soil environments [112], and climate conditions can all contribute to disease development. It is desirable to develop prediction models with the capacity to integrate all of the information to make precise forecasting. Recently, AI and big data methods have been used to study disease factors and to manage disease in the field [166].

DL has been developed in integrating environmental conditions to predict disease outbursts in the field. For example, Xiao et al. (2018) developed an LSTM model to predict the outbreak of pests (bollworm, whitefly, and jassid) and fungal disease (leaf blight) for cotton under consecutive weather conditions (e.g., maximum temperature, minimum temperature, relative humidity, and rainfall) [43]. The results indicated that the LSTM model reached a very high prediction performance, and outperformed the other three models (KNN, SVC, and RF) that did not take sequence information into account. Combining CNN and LSTM, the model used to predict the outbreak of *P. dispersa* (leaf rust) on wheat was also developed by Pryzant et al. (2017) [167]. In this study, 8554 observations, each of which include geology information (latitude and longitude), gross primary productivity, land surface temperature, and remote sensing data on the surface, were used to predict disease severity (on the stem, stripe, and leaf). The prediction results (with an accuracy of 76.53%) were promising for expansion with other diseases. To predict wheat yellow rust breaks, Xu et al. (2018) proposed an RNN-based model of a spatial–temporal recurrent neural network (STRNN) for agricultural emergency management [168]. In this model, time-series data (remotely sensed data) and non-time-series data (climatic, topographic, and soil data) were integrated into the spatial–temporal data vectors in STRNN. The performance of this model, by considering the data dependencies between different time points, outperformed the best baseline models of RF by 30.20%, 23.31%, and 44.52% in the evaluation indexes of MAE, MAPE, and RMSE, respectively. The outperformance of the RNN model is understandable, as baseline models ignore the fact that the dynamic features such as weather for a time span are complex multi-dimensional time series data. The high relevance of dependence between different time points is not appropriate to be fed into fixed-size networks for model training, and the dependence between input units across time needs adequate attention. With two years of data on climate and biotic stresses, we developed a DL model to forecast blackleg disease outbreak, and the accuracy reached 66% (manuscript submitted).

Although diverse factors have been collected to forecast disease in commercial fields, microbe data are surprisingly ignored in disease forecasting, given the pathogen’s importance for disease development in the field. For example, pathogen race and the avr gene profile, through gene-to-gene interactions with plant R gene, are very critical for the development of plant diseases [169]. In addition, both the qualitative and quantitative effects will determine the level of resistance. Along with continuously reducing the cost of sequencing, collecting information on microbe communities and plant resistance will become more practical. Given the successful integration of diverse sources of data in pioneering DL-based disease forecasting studies, full usage of microbe information such as microbe races and genotypes is promising, as well as functions to predict disease with high accuracy.

## 5. Future Perspectives

As a science driven by genomic and amplicon sequencing, and phenomic databases, phytomicrobiomes are an ideal application for AI. Although there are a growing number of studies that focus on the phytomicrobiome, there are still relatively few publicly available datasets that can be used for training and validating AI models. This can make it challenging to develop accurate predictive models or to identify patterns in phytomicrobiome data. Challenges also come from sample collection and processing. Collecting and processing phytomicrobiome samples can be difficult and time-consuming, as it often involves separating plant and microbial tissues and removing contaminants. This can make it challenging to obtain large datasets that are representative of the phytomicrobiome in different environments, and to collect similar data across studies. To address data shortages in phytomicrobiome research, future efforts are needed to increase the number of publicly available datasets and to develop standardized protocols for sample collection and processing. Finally, advances in ML techniques, such as transfer learning, active learning, and data augmentation can help to address the challenges related to the limited availability of datasets and the need for accurate predictive models.

Integrating phytomicrobiome data with other omics data can provide a more comprehensive understanding of the interactions between plants and their associated microbial communities. For example, by integrating phytomicrobiome data with transcriptomic data, it is possible to identify genes that are differentially expressed in response to specific microbial taxa or environmental conditions. By integrating phytomicrobiome data with metabolomics data, it is possible to identify metabolic pathways that are influenced by specific microbial taxa or host growth conditions. Other omics data such as proteomic data and host genomic data can also be integrated to understand plant–microbe interactions. Integrating phytomicrobiome data with other omics data can be challenging, as it often involves analyzing large, complex datasets and integrating data from multiple sources. However, advances in ML techniques are making it increasingly possible to integrate diverse omics data and to identify patterns and relationships that would be difficult to detect using individual datasets alone.

Although there are AI tools designed for specific scientific purposes, there is room for improvement to make these tools more easily accessible. This is a likely natural progression in the tools and has been observed in other science disciplines, such as genomics, which was previously conducted using a few niche specific Linux command line tools, and is now achievable on a wide set of Windows-based all-in-one software with graphical user interfaces (i.e., CLC Genomics WorkBench, Geneious, and others). Recently, OpenAI has released public and user-centric AI tools for creating and interacting with data, text, and images, which have become immensely popular amongst the general public; thus, the transition of AI tools is already underway. Developing improved AI tools for phytomicrobiome analysis will also be urgent, in order to make the process more accessible to researchers who may not have a background in AI. For example, developing user-friendly software that allows researchers to easily upload and analyze their data can help to make AI more accessible. This software could include pre-built AI models and user-friendly interfaces for data pre-processing and model training. Visualization tools could allow researchers to visualize their data, and the results of their AI models can be more intuitive. These tools could include interactive graphs, heat maps, and other visualizations that allow researchers to explore their data in real time. To provide a clear explanation of how to use AI tools, tutorials and documentation should include step-by-step guides for data pre-processing, model training, and model evaluation, as well as examples for specific research applications. Cloud-based solutions are also waiting for establishment, to provide researchers with access to powerful computing resources without the need for expensive hardware or software. By developing cloud-based AI solutions, researchers can easily access the tools and resources they need to analyze their data.

## 6. Conclusions

Deciphering the interplay between plant microbiomes and crop production is challenging. Data mining can be used to extract taxonomic composition, gene functions, and associations between plant microbiomes and host plant phenotypes. AI, boosted by ML and DL advancements, is being explored as a solution to meet some of the challenges associated with data size and complexity when considering the plant microbiome in cropping systems. The diverse sources of ‘omics’ data present a good opportunity for AI to predict agronomic outcomes by combining information from plant-associated microbes, plant genomes, and soil properties, as well as climate conditions. AI is advantageous in analyzing high-dimensional variables, processing data with flexible architectures, and extracting intrinsic patterns from phytomicrobiome data without complex and skill-required feature selection. Such abilities make AI very suitable for speculating key information from microbiome data, which have high-dimensionality and incomplete databases. By predicting the plant growth status in the field, AI can also integrate plant genomes through genome prediction techniques. Plant traits such as disease resistance vary in the field because of multi-lateral interactions among plant genomes, the dynamics of the pathogen population, microbe composition, and interactions across environments. Genome prediction with AI can take full advantage of the available data to predict the breeding values of individuals while avoiding the high cost of phenotyping all individuals. High-throughput plant phenotyping in the field also makes it possible to gather large amounts of data that can be integrated into the models and that can predict plant phenotypes. By combining diverse sets of data on microbiomes, plant pathogens, phenomics, crop genomics, and climate data, it is also possible to predict crop performance and the occurrences of pathogen outbreaks. While the application of AI in the above-mentioned areas may still be in its infancy, many application-specific tools are now being put into practice; they have been demonstrated to be remarkably efficient and accurate, and they may be ushering in a new era of AI-driven crop research.

## Figures and Tables

**Figure 1 plants-12-01852-f001:**
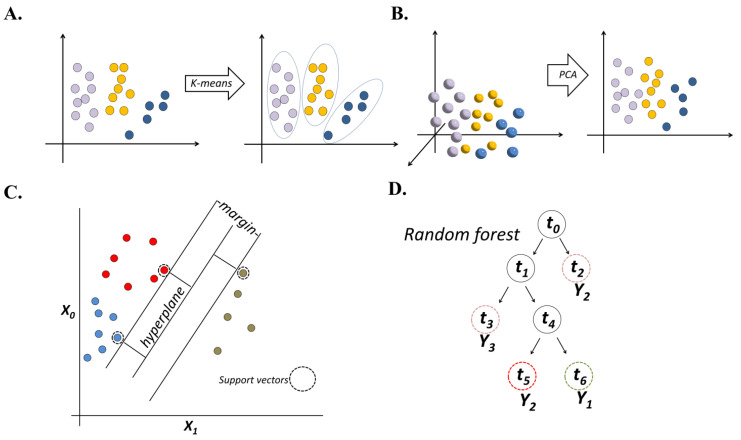
Illustration of several machine learning (ML) methods. Balls with the same color represent samples from the same class, and different colors were used to represent different classes. (**A**) K-means. (**B**) Principal component analysis (PCA) method. (**C**) Support vector machines (SVMs) method. (**D**) Random forest (RF) method.

**Figure 2 plants-12-01852-f002:**
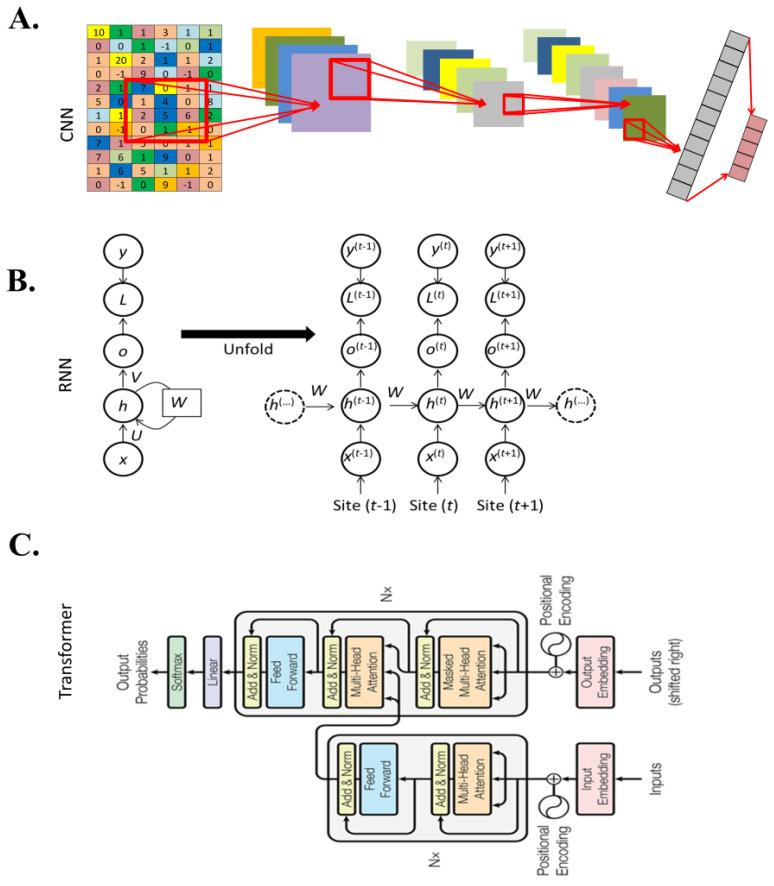
Illustration of major deep learning (DL) methods. (**A**) Illustration of the convolutional neural network (CNN) model. In a convolutional layer, the convolutional operation with a pre-defined window (height ×width), and stride is conducted across the input matrix. The convolutional layers act as automatic feature extractors from the input images. (**B**) Illustration of recurrent neural network (RNN) models. RNN processes the inputs one-by-one, progressively based on the input DNA sequence. The output from the previous layer(s) is processed at the current neuron, together with the current input, making the current neuron produce output with the consideration of “memories”. (**C**) Illustration of self-attention model [41]. The self-attention mechanism takes into account each nucleotide in a DNA sequence at the same time and decides which ones are important by attributing different weights.

## Data Availability

Not applicable.

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
