# Peer review of "Artificial Intelligence: A Promising Tool in Exploring the Phytomicrobiome in Managing Disease and Promoting Plant Health"

_plants, 2023, doi:10.3390/plants12091852_

Round 1

Reviewer 1 Report

Dear Authors,

Great work! My only comment is related to phrasing. Please double check for typos and phrasing.

Best regards,

R.

Author Response

Thank you very much for your advice! We have gone through the phrasing and typos carefully, and we have made corresponding changes across the manuscript.

Reviewer 2 Report

The review by Zhao et al. is dedicated to the use of artificial intelligence approaches to dissect metagenomic information about microbial communities and prediction of different scenarios of plant-microbiome interaction. This is very timely review because it demonstrates the state of the art of the very hot research direction on the intersection of the two blooming knowledge areas (metagenomics and AI/ML). This review highlights how the scientists working in the field of symbiology and biology of complex plant-microbial interaction can benefit from the novel mathematical methods of data analysis. Many convincing cases are presented providing reader with valuable information on the specific approaches to implementing of the AI-based approaches to (meta)genome mining and making prediction about the important biological traits. The manuscript is well written. Certainly, I would love to see this review published since it makes much value to the readership of the journal.

I have but a few small suggestions:

LL362-364: ITS1-5.8S rRNA-ITS2 loci are useful as DNA barcodes not only for fungi but for other eukaryotes as well, right?

L421: mthods -> methods

Section 4.2 is essentially about image processing, not about mining the genomic information and therefore can be painlessly omitted or merged with other section.

The introduction of the ML methods is fine but it is desirable to see some more detailed demonstrations of the under-the-hood mechanics of the AI/ML-based systems that were successfully used for solving typical tasks of metagenomic studies.

Author Response

Q1: LL362-364: ITS1-5.8S rRNA-ITS2 loci are useful as DNA barcodes not only for fungi but for other eukaryotes as well, right?

A1: Thank you for providing these great comments!

For the ITS loci, the ITS1-5.8S rRNA-ITS2 loci can be found in many eukaryotic organisms, but their usefulness as molecular markers can vary depending on the group being studied.

For example, the ITS region has been extensively studied in fungi and is considered a highly reliable marker for fungal phylogenetics due to its high sequence variability and conserved flanking regions. Similarly, the ITS region is also commonly used for plant phylogenetics.

However, the ITS region may not be as useful for some animal groups because of its more limited sequence variability in those organisms. In these cases, other molecular markers may be more appropriate for phylogenetic analysis.

Therefore, while the ITS1-5.8S rRNA-ITS2 loci can be found in many eukaryotic organisms, their effectiveness as a molecular marker can depend on the specific group of organisms being studied.

We have revised the statement to clarify that it is in eukaryotes and is useful for fungal phylogentics (line 363).

---

Q2: L421: mthods -> methods

A2: Thank you for pointing this out, we have made the correction.

---

Q3: Section 4.2 is essentially about image processing, not about mining the genomic information and therefore can be painlessly omitted or merged with other section.

A3: Thank you for this great suggestion. Now we have merged the sections of 4.2 and 4.3 into 4.2. High-throughput Plant Disease Phenotyping and In-field Plant Disease Forecasting.

---

Q4: The introduction of the ML methods is fine but it is desirable to see some more detailed demonstrations of the under-the-hood mechanics of the AI/ML-based systems that were successfully used for solving typical tasks of metagenomic studies.

A4: Thanks for this great suggestion! We prepared a list (Table 1) of AI applications with these AI/ML models. As the list is relatively long, we thought it might be better to present these studies in a table, rather than integrate the information into the main text. In Table 1, major information such as the used AI model, research question, as well as sequence type was listed. We hoped that by doing this, readers would be able to access the referenced studies to obtain specific information that aligns with their interests, and it would help ensure that this manuscript is well-organized and free onerous detail that may be less applicable to a broad audience.

Reviewer 3 Report

The submitted manuscript is designed as a literature review, which summarizes the acquired scientific knowledge to summarize the information achieved so far. It is a pity that the works of the authors of the manuscript are not listed in the overview of the literature used. Do the authors engage in this area of research in their scientific activity? If so, it would be appropriate to supplement the publication with your own experimental data. The review summarizes the latest scientific knowledge compared to older results. From a didactic point of view, the mentioned manuscript is an asset. It can also be considered successful as a source of relevant scientific information. There are two figures and one table in the text. I recommend adding this, as the text will gain quality. The text is written carefully, with a logical structure and continuity. I recommend correcting and unifying the overview of used literary sources.

Author Response

Thank you for the great suggestion that to list our current work related to this manuscript.

A1: Yes, we actively engaged in the areas of 1) deep learning based genomic selection for barley FHB resistance, 2) in-field high-throughput and precise disease phenotyping, and 3) AI-based disease forecasting using climate and biological factors. For these areas, we have developed models (optimizing parameters), published manuscripts (cited in our review manuscript), a research manuscript under review etc. We are also actively conducting phytomicrobiome analysis on FHB disease, and AI-based analysis could be the next step.

Our cited work in this review manuscript is (line 722):

Jubair S.; Tucker J.R.; Henderson N.; Hiebert C.W.; Badea A.; Domaratzki M.; Fernando W. GPTransformer: a transformer-based deep learning method for predicting Fusarium related traits in barley. Front. Plant Sci. 2021, 12:2984.

We also mentioned our work on high-throughput disease identification and disease forecasting in section 4.2 (line 849-852 for disease identification, and line 912-914 for disease forecasting). We also presented the information below for your reference.

Line 849-852: Our lab is also developing DL models to effectively rate canola blackleg disease and flea beetle bites on seedling cotyledon.

Line 912-914: With two years of data of climate and biotic stresses, we developed a DL model to forecast blackleg disease outbreak and the accuracy reached 66% (our submitted manuscript).

Q2: There are two figures and one table in the text. I recommend adding this, as the text will gain quality. The text is written carefully, with a logical structure and continuity. I recommend correcting and unifying the overview of used literary sources.

--

 A2: Your suggestion that recommending us to integrate the content in Table 1 and figures into the text is excellent! We prepared a list (Table 1) of AI applications with these AI/ML models. As the list is relatively long, we thought it might be better to present these studies in a table, rather than integrate the information into the main text. In Table 1, major information such as the used AI model, research question, as well as sequence type was listed. We hoped that by doing this, readers would be able to access the referenced study to obtain specific information that aligns with their interests, and it would help ensure that this manuscript is well-organized and free of onerous detail that may be less applicable to a broad audience. For figure information, we briefly explained the model principle in section 2 and focus on real application examples in section 3.

Reviewer 4 Report

In this review, the aim was to provide a brief introduction to artificial intelligence (AI) techniques and introduce how AI has been applied in areas of microbiome sequencing taxonomy, the functional annotation for  microbiome sequences, associating microbiome community to host traits, designing synthetic communities, genomic selection, field phenotyping, and disease forecasting. The review conatins som einteresting aspeczs of the topic. The study can be considered after suitable revisions.

Abstract: Please give a final conculding or future aspects sentence at the end of the Abstract.

Number of keyword are too much. Please reduce it to 5-8.

L178: Figure 1. Please give abbreviations in full in the title (PCA, SVM)

L284: Figure 2. Please give abbreviations in full in the title (DL, CNN)

Please give a sperate section of Future ascpects. At around L899.

I missed authors constribution section.

References: Formating in titles, journal names are inconsistent. 

Author Response

Reviewer 4:

In this review, the aim was to provide a brief introduction to artificial intelligence (AI) techniques and introduce how AI has been applied in areas of microbiome sequencing taxonomy, the functional annotation for  microbiome sequences, associating microbiome community to host traits, designing synthetic communities, genomic selection, field phenotyping, and disease forecasting. The review conatins som einteresting aspeczs of the topic. The study can be considered after suitable revisions.

--

Q1: Abstract: Please give a final conculding or future aspects sentence at the end of the Abstract.

A1: Thank you for this conclusion. We added a sentence at the end of the Abstract.

--

Q2: Number of keyword are too much. Please reduce it to 5-8.

A2: Thank you. We have reduced to six keywords.

--

Q3: L178: Figure 1. Please give abbreviations in full in the title (PCA, SVM)

L284: Figure 2. Please give abbreviations in full in the title (DL, CNN)

A3: Thank you. We have provided the full name of these models.

--

Q4: Please give a sperate section of Future ascpects. At around L899.

A4: Thank you for this great suggestion. We added a separate section (section 5) for future aspects (line 930-998). We also provide this section here for your convenience:

“5. Future Perspectives

As a science driven by genomic and amplicon sequencing and phenomic databases, phytomicrobiomes  is an ideal application for AI. Although there are a growing number of studies that focus on the phytomicrobiome, there are still relatively few publicly available datasets that can be used for training and validating AI models. This can make it challenging to develop accurate predictive models or to identify patterns in phytomicrobiome data. The challenges also come from sample collection and processing. Collecting and processing phytomicrobiome samples can be difficult and time-consuming, as it often involves separating plant and microbial tissues and removing contaminants. This can make it challenging to obtain large datasets that are representative of the phytomicrobiome in different environments, and to collect similar data across studies. To address data shortage in phytomicrobiome research, future efforts are needed to increase the number of publicly available datasets and to develop standardized protocols for sample collection and processing. Finally, advances in ML techniques, such as transfer learning, active learning, and data augmentation, can help to address the challenges related to limited availability of datasets and the need for accurate predictive models.

Integrating phytomicrobiome data with other omics data can provide a more comprehensive understanding of the interactions between plants and their associated microbial communities. For example, by integrating phytomicrobiome data with transcriptomic data, it is possible to identify genes that are differentially expressed in response to specific microbial taxa or environmental conditions. By integrating phytomicrobiome data with metabolomics data, it is possible to identify metabolic pathways that are influenced by specific microbial taxa or host growth conditions. Other omics data such as proteomic data and host genomic data can also be integrated to understand plant-microbe interactions. Integrating phytomicrobiome data with other omics data can be challenging, as it often involves analyzing large, complex datasets and integrating data from multiple sources. However, advances in ML techniques are making it increasingly possible to integrate diverse omics data and to identify patterns and relationships that would be difficult to detect using individual datasets alone.

While AI tools for targeted scientific applications are available, improvements could be made to make these tools more accessible. This is a likely natural progression in the tools and has been observed in other science disciplines, such as genomics, which was previously conducted using a few niche specific Linux commandline tools, and is now achievable on a wide set of Windows based all-in-one software with graphic user interfaces (i.e. CLC Genomics WorkBench, Geneious, and others). Recently, OpenAI has released public and user-centric AI tools for creating and interacting with data, text, and images, which have become immensely popular amongst the general public; thus, the transition of AI tools is already underway.  Developing improved AI tools for phytomicrobiome analysis will also be urgent to make the process more accessible to researchers who may not have a background in AI. For example, developing user-friendly software that allows researchers to easily upload and analyze their data can help to make AI more accessible. This software could include pre-built AI models and user-friendly interfaces for data preprocessing and model training. Visualization tools could allow researchers to visualize their data and the results of their AI models can be more intuitive. These tools could include interactive graphs, heat maps, and other visualizations that allow researchers to explore their data in real time. To provide a clear explanation of how to use AI tools, tutorials and documentation are necessary to include step-by-step guides for data preprocessing, model training, and model evaluation, as well as examples for specific research applications. Cloud-based solutions are also waiting for the establishment to provide researchers with access to powerful computing resources without the need for expensive hardware or software. By developing cloud-based AI solutions, researchers can easily access the tools and resources they need to analyze their data.”

--

Q5: I missed authors constribution section.

A5: Thank you again! We added this section (line 1036-1039).

---

Q6: References: Formating in titles, journal names are inconsistent. 

A6: Thank you for the careful checking. We have gone through reference formats and made several changes.
